# Constrained Monotonic Neural Networks

## Abstract

Deep neural networks are becoming increasingly popular in approximating arbitrary functions from noisy data. But wider adoption is being hindered by the need to explain such models and to impose additional constraints on them. Monotonicity constraint is one of the most requested properties in real-world scenarios and is the focus of this paper. One of the oldest ways to construct a monotonic fully connected neural network is to constrain its weights to be non-negative while employing a monotonic activation function. Unfortunately, this construction does not work with popular non-saturated activation functions such as ReLU, ELU, SELU etc, as it can only approximate convex functions. We show this shortcoming can be fixed by employing the original activation function for a part of the neurons in the layer, and employing its point reflection for the other part. Our experiments show this approach of building monotonic deep neural networks have matching or better accuracy when compared to other state-of-the-art methods such as deep lattice networks or monotonic networks obtained by heuristic regularization. This method is the simplest one in the sense of having the least number of parameters, not requiring any modifications to the learning procedure or post-learning steps. Finally, we give a proof it can approximate any continuous monotone function on a compact subset of $\mathbb{R}^n$.

## 1 Introduction

Deep Learning has witnessed widespread adoption in many critical real-world domains such as finance, healthcare, etc [21]. Predictive models built using deep neural networks have been shown to have high accuracy. Incorporating prior knowledge such as monotonicity in trained models help in improving the performance and generalization ability of the trained models [25, 9]. The introduction of structural biases such as monotonicity makes models also more data-efficient, enabling a leap in predictive power on smaller datasets [42]. Apart from the requirements of having models with high accuracy, there is also a need for transparency and interpretability, and monotonicity helps in partially achieving the above requirements [14]. Due to legal, ethical and/or safety concerns, monotonicity of predictive models with respect to some input or all the inputs is required in numerous domains such as financial (house pricing, credit scoring, insurance risk), healthcare (medical diagnosis, patient medication) and legal (criminal sentencing) to list just a few. All other things being equal, a larger house should be deemed more valuable, a bank's clients with higher income should be eligible for a larger loan [36], and an offender with a longer crime history should be predicted as more likely to commit another crime [32], etc. A model without such a monotonic property would not, and certainly should not, be trusted by society to provide a basis for such important decisions. However, the monotonicity of deep learning models is not a guaranteed property even when trained on monotonic data, let alone when training on noisy data typically used in practice.

Although monotonicity is an important and increasingly often even required property, there is no simple or easy method to enforce it. It has been an active area of research and the existing methods on the subject can be broadly categorized into two types:

Submitted to 36th Conference on Neural Information Processing Systems (NeurIPS 2022). Do not distribute.

1. Monotonic architectures by construction: This line of research focuses on neural architectures guaranteeing monotonicity by construction [2, 37, 8, 24, 43].

2. Monotonicity by regularization: This line of research focuses on enforcing monotonicity in neural networks during training by employing a modified loss function or a heuristic regularization term [38, 13].

We give a more detailed account of the existing methods in the next section.

The simplest method to achieve monotonicity by construction is to constrain the weights of the fully connected neural network to have only non-negative (for non-decreasing variables) or only non-positive values (for non-ascending) variables when used in conjunction with a monotonic activation function, a technique known for almost 30 years [2]. However, this method does not work well in practice. When used in conjunction with saturated (bounded) activation functions such as the sigmoid and hyperbolic tangent, these models are difficult to train, i.e. they do not converge to a good solution. On the other hand, when used with non-saturated (unbounded) convex activation functions such as ReLU [26], the resulting models are always convex [22], severely limiting the applicability of the method in practice.

The main contribution of this paper is a simple modification of the method above which, in conjunction with non-saturated activation functions, is capable of approximating non-convex functions as well: if both concave and convex monotone activation functions are used in a neural network with constrained weights, it regains the ability to approximate monotone continuous functions that are either convex, concave or nothing of the above.

There are several possible ways to satisfy this condition, with the simplest one using both an activation function $\breve{\rho}$ and its point reflection with respect to point $(0, 0)$ defined as:

$$\hat{\rho}(x) = -\breve{\rho}(-x)$$

Assuming the original activation function $\breve{\rho}$ is both monotonic and convex, the properties holding for ReLU, ELU, SELU, and Softplus, the proposed modification uses the original activation function $\breve{\rho}$ on a part of the neurons in the network and its point reflection $\hat{\rho}$ on the rest of the neurons.

The resulting model is guaranteed to be monotonic, can be used in conjunction with any convex monotonic non-saturated activation function, doesn't have any additional parameters compared to a non-monotonic fully-connected network for the same task, and can be trained without any additional requirements on the learning procedure. Experimental results show it is exceeding the performance of all other state-of-the-art methods, all while being both simpler (in the number of parameters) and easier to train.

Our contributions can be summarized as follows:

1. We propose a modification to an existing constricted neural network layer enabling it to model non-convex functions when used with non-saturated monotone convex activation functions such as ReLU, ELU, SELU, and alike.

2. We perform comparisons with other recent works and show that our proposed novel neural network block can yield comparable and in some cases better results than the previous state-of-the-art and with significantly fewer parameters.

3. We prove the universal approximation property for the ReLU activation function, showing that the proposed architecture can approximate any monotone continuous function on a compact subset of $\mathbb{R}^n$.

## 2   Related work

Before dwelling on the methods employed to build monotonic models, we give an overview of activation functions as they form a crucial part of our work.

### 2.1   Activation functions

Right from its inception in perceptron [31], non-linear activation functions have historically been one of the most important components of neural networks. Previously, the saturated functions such

as the sigmoid [33], the hyperbolic tangent [27], and its variants were the most common choice of activation functions. Currently, one of the most important factors for state-of-the-art results accomplished by modern neural networks is the use of non-saturated activation functions. The use of *Rectified Linear Unit* (ReLU) [26, 12] as activation function was instrumental in achieving good performance in newer architectures. The ReLU has since become a defacto choice of activation in most practical implementations and continues to be widely used because of its advantages such as simple computation, representational sparsity, and linearity. Later, a number of activation functions were proposed to deal with solving problems of dead neurons and aid in faster convergence [23], [6] [15], [45], [16], [30], [20].

The idea of using both the original activation function and its point reflection in the same layer has been proposed in [35] where both outputs of ReLU and the negative value of its point reflection were used in the construction of *concatenated ReLU* (CReLU) activation function. The proposed modification outputs two values instead of one and therefore increases the number of parameters. In [44], the authors propose *negative concatenated ReLU* (NCReLU) flip the sign and use the point reflection directly. Notice that the proposed architectural change could be applied to other non-saturated, monotonic activation functions as well, but with an unknown impact on their performance.

In [10], the authors propose *bipolar ReLU* which consists of using ReLU on the half of the neurons in the layer and the point reflection of ReLU on the other half. The same construction could be used with other ReLU family activation functions as well. However, the focus of their work was to alleviate the need of normalizing layers and improve the performance of deep vanilla recurrent neural networks (RNNs) by using *bipolar ReLU*.

Both NCReLU and bipolar ReLU could have been used in the construction of a constrained neural network capable of representing non-convex functions, but to the best of our knowledge, they have not been so far.

## 2.2 Monotonicity by construction

Apart from the approaches mentioned in the introduction (section 1), another approach to building monotonic neural architecture is Min-Max networks where monotonic linear embedding and max-min-pooling are used [37]. In [8], authors generalized this approach to handle functions that are partially monotonic and proved that the resulting networks have the universal approximation property. However, such networks are very difficult to train and not used in practice. Their construction does not allow for replacement with other activation functions.

Deep lattice networks (DLN) [43] use a combination of linear calibrators and lattices [24] for learning monotonic functions. This is the most widely used method in practice today, but not without its limits. Lattices are structurally rigid thereby restricting the hypothesis space significantly. Also, DLN requires a very large number of parameters to obtain good performance.

Given a model with a convex output function, it is possible to use backpropagation [34] to make a monotonic model by computing the derivation of the output function. One simple way to construct a convex function is to use an unsaturated monotonic activation function in a fully connected layer as mentioned above, but we could also use a more elaborate architecture such as the input convex neural networks [1]. Although possible, these kinds of constructions are computationally more complex than the simple solution proposed here.

## 2.3 Monotonicity by regularization

In the second category, the research works focus on enforcing monotonicity during the training process by modifying the loss function or by adding a regularization term.

In [38], the authors propose a modified loss function that penalizes the non-monotonicity of the model. The algorithm models the input distribution as a joint Gaussian estimated from the training data and samples random pairs of monotonic points that are added to the training data. In [13], the authors propose a point-wise loss function that acts as a soft monotonicity constraint. These methods are straightforward to implement and can be used with any neural network architecture. However, these methods do not guarantee the monotonicity of the trained model.

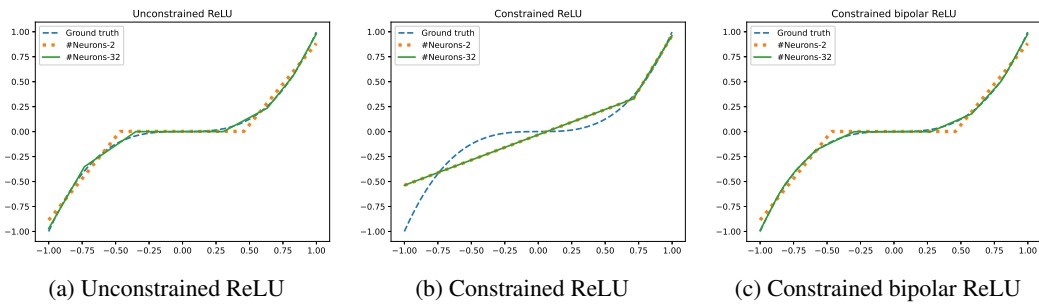



(a) Unconstrained ReLU      (b) Constrained ReLU      (c) Constrained bipolar ReLU



Figure 1: Approximations of the cubic function $f(x) = x^3$

Recently, there is an increasing number of proposed methods to certify or verify monotonicity obtained by regularization methods. In [22], the authors propose an optimization-based technique for mathematically verifying, or rejecting, the monotonicity of an arbitrary piece-wise linear (e.g., ReLU) neural network. The method consists of transforming the monotonicity verification problem into a mixed integer linear programming (MILP) problem that can be solved using an off-the-shelf MILP solver.

In [39], the authors propose an approach that finds counterexamples (defined as the pair of points where the monotonicity constraint is violated) by employing satisfiability modulo theories (SMT) solver [3]. To satisfy the monotonicity constraints, these counterexamples are included in the training data with adjustments to their target values to enforce the next iterations of the model to be monotonic.

Both methods [22, 39] have been shown to support ReLU as the activation function only and there is no obvious way how to extend them to other activation functions. More precisely, they rely on piece-wise linearity of ReLU to work, the property not satisfied by other variants such as ELU, SELU, GELU, etc. Last but not least, the procedure for certifying/verifying using MILP or SMT solvers is computationally very costly. These approaches also require multiple reruns or iterations to arrive at certified/verified monotonic networks.

## 3   Constrained neural networks

Most of the commonly used activation functions such as ReLU, ELU, SELU, etc. are monotonically increasing, convex, non-polynomial functions. When used in a fully-connected, feed-forward neural network with at least one hidden layer and with unconstrained weights, they can approximate any continuous function on a compact subset. The simplest way to construct a monotonic neural network is to constrain its weights when used in conjunction with a monotone activation function. However, when the activation function is convex as well, the constrained neural network is not able to approximate non-convex functions.

To better illustrate this, and to propose a simple solution in this particular example, we refer the readers to Figure 1 where the goal is to approximate a simple cubic function $x^3$ using a neural network with a single hidden layer with either 2 or 32 neurons and with ReLU activation. A cubic function is apt for our illustration since it is concave in the considered interval $[-1, 0]$ and convex in the interval $[0, -1]$:

Fig. 1a. An unconstrained ReLU network with $n$ neurons can approximate both concave and convex segments of the cubic function using at most $n + 1$ piecewise linear segments. Increasing the number of neurons will provide a better fit with the function being approximated. Notice that even though the cubic function is monotone, there is no guarantee that the trained model will be monotone as well.

Fig. 1b. If we constrain the weights of the network to be non-negative while still employing ReLU activation, the resulting model is monotone and convex. We can no longer approximate non-convex segments such as the cubic function on $[-1, 0]$ in the figure, and increasing the number of neurons from 2 to 32 does not yield any significant improvement in the approximation.

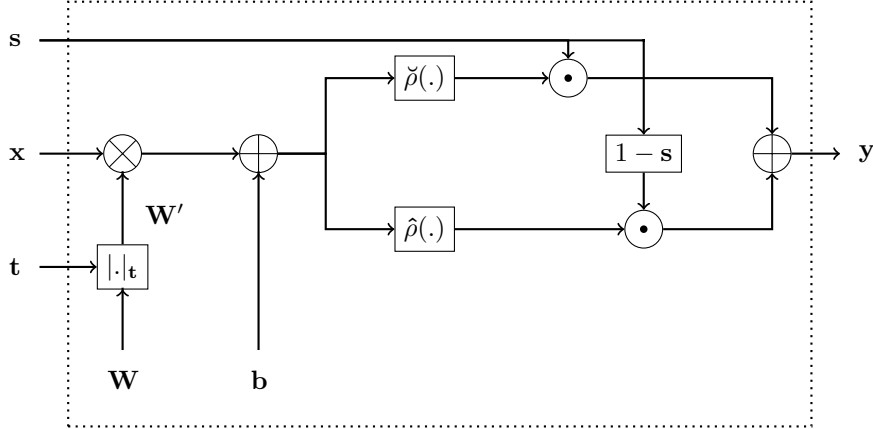

Figure 2: Proposed Monotonic Dense Unit or Constrained Monotone Fully Connected Layer

Fig. 1c. The proposed solution uses both convex and concave activation functions in the hidden layer, in this case, bipolar ReLU, to gain the ability to model non-convex, monotone continuous functions. Notice that increasing the number of neurons increases the number of piecewise linear segments to approximate the cubic function. The resulting network is monotone by construction even when trained on noisy data.

The schematic block diagram of our proposed solution (which we refer to as Constrained Monotone Fully Connected Layer or Monotonic Dense Unit interchangeably) is shown in the figure Fig. 2. The individual components of the proposed solution are defined and described in the subsequent subsection.

## 3.1 Constrained monotone fully connected layer

A function $f$ is partially monotone with respect to its parameter $x_i$ if $\dfrac{\partial f}{\partial x_i}(x_1, \ldots, x_n)$ is non-negative or non-negative for all $x_1, \ldots, x_n$. A set $S \subseteq R$ is called compact if every sequence in $S$ has a subsequence that converges to a point in $S$. One can easily show that closed intervals $[a, b]$ are compact, and compact sets can be thought of as generalizations of such closed bounded intervals.

Our construction is preconditioned on a priori knowledge of (partial) monotonicity of a multivariate, multidimensional function $f$. Let $f : K \mapsto \mathbb{R}^m$ be defined on a compact segment $K \subseteq \mathbb{R}^n$. Then we define its $n$-dimensional *monotonicity indicator vector* $\mathbf{t}$ element wise as follows:

$$
t_i = \begin{cases}
1 & \text{if } \dfrac{\partial f(\mathbf{x})_j}{\partial x_i} \geq 0 \text{ for each } j \in \{1, \ldots, m\} \\
-1 & \text{if } \dfrac{\partial f(\mathbf{x})_j}{\partial x_i} \leq 0 \text{ for each } j \in \{1, \ldots, m\} \\
0 & \text{otherwise}
\end{cases}
\tag{1}
$$

Given an $(n \times m)$-dimensional matrix $\mathbf{W}$ and $n$-dimensional monotonicity indicator vector $\mathbf{t}$, we define the operation $|.|_t$ assigning an $(n \times m)$-dimensional matrix $\mathbf{W}' = |\mathbf{W}|_t$ to $\mathbf{W}$ as follows:

$$
w'_{i,j} = \begin{cases}
|w_{i,j}| & \text{if } t_i = 1 \\
-|w_{i,j}| & \text{if } t_i = -1 \\
w_{i,j} & \text{otherwise}
\end{cases}
\tag{2}
$$

**Definition 1** (Constrained linear layer). *Let $\mathbf{W} \in \mathbb{R}^{n \times m}$, $\mathbf{t} \in \mathbb{R}^n$, $\mathbf{x} \in \mathbb{R}^n$ and $\mathbf{b} \in \mathbb{R}^n$. The output $\mathbf{h} \in \mathbb{R}^m$ of the* constrained linear layer *with monotonicity indicator vector $\mathbf{t}$, weights $\mathbf{W}$, biases $\mathbf{b}$ and input $\mathbf{x}$ is:*

$$
\mathbf{h} = |\mathbf{W}|_{\mathbf{t}} \cdot \mathbf{x} + \mathbf{b}
\tag{3}
$$

**Lemma 1.** *For each $i \in \{1, \ldots, n\}$ and $j \in \{1, \ldots, m\}$ we have:*

- *if $t_i = 1$, then $\dfrac{\partial h_j}{\partial x_i} \geq 0$, and*

- *if $t_i = -1$, then $\dfrac{\partial h_j}{\partial x_i} \leq 0$.*

Let $\check{\rho}$ be a monotonically increasing *convex* function such as ReLU, ELU, etc. Then its point reflection $\hat{\rho}$ around the origin is:

$$\hat{\rho}(x) = -\check{\rho}(-x) \tag{4}$$

Notice that $\hat{\rho}$ is a monotonically increasing *concave* function.

**Definition 2** (Combined activation function). *Given a monotonically increasing* convex *function $\check{\rho}$, its point reflection $\hat{\rho}$ and $m$-dimensional* activation selector vector $\mathbf{s} \in [0,1]^m$*, the output of the* combined activation function $\rho^{\mathbf{s}}$ *is a weighted sum of $\check{\rho}$ and $\hat{\rho}$:*

$$\rho^{\mathbf{s}}(\mathbf{h}) = \mathbf{s} \odot \check{\rho}(\mathbf{h}) + (\mathbf{1} - \mathbf{s}) \odot \hat{\rho}(\mathbf{h}) \tag{5}$$

*where $\odot$ is the element-wise (Hadamard) product and $\mathbf{1}$ is $m$-dimensional vector with all elements equal to 1.*

**Lemma 2.** *Let $\mathbf{y} = \rho^{\mathbf{s}}(\mathbf{h})$. Then for each $j \in \{1, \ldots, m\}$ we have $\dfrac{\partial y_j}{\partial h_j} \geq 0$. Moreover*

- *if $\mathbf{s} = \mathbf{1}$, then $\rho_j^{\mathbf{s}}$ is convex; and*

- *if $\mathbf{s} = \mathbf{0}$, then $\rho_j^{\mathbf{s}}$ is concave.*

**Definition 3** (Monotone constrained fully connected layer). *Let $n$ and $m$ be positive natural numbers, $\check{\rho}$ a monotonically increasing* convex *function, $\mathbf{t}$ an $n$-dimensional monotonicity selector vector and $\mathbf{s}$ an $m$-dimensional activation selector vector. Then the* monotone constrained fully connected layer *with $m$ neurons is a tuple $(\check{\rho}, \mathbf{t}, \mathbf{s})$, denoted $\mathcal{MFC}_{\check{\rho},\mathbf{s},\mathbf{t}}$.*

*Moreover, let weights $\mathbf{W}$ be an $(n \times m)$-dimensional matrix, biases $\mathbf{b}$ an $m$-dimensional vector and input $x$ an $n$-dimensional vector. The* output function *of the monotone constrained fully connected layer, denoted $\mathcal{MFC}_{\check{\rho},\mathbf{s},\mathbf{t}}^{\mathbf{W},\mathbf{b}}(\mathbf{x})$, assigns an $m$-dimensional vector $\mathbf{y}$ to $(\mathbf{W}, \mathbf{b}, \mathbf{x})$ as follows:*

$$\mathbf{y} = \mathcal{MFC}_{\check{\rho},\mathbf{s},\mathbf{t}}^{\mathbf{W},\mathbf{b}}(\mathbf{x}) = \rho^{\mathbf{s}}\left(|\mathbf{W}|_{\mathbf{t}} \cdot \mathbf{x} + \mathbf{b}\right) \tag{6}$$

Notice that $\mathcal{MFC}_{\check{\rho},\mathbf{s},\mathbf{t}}$ determines an architecture of the layer, while $\mathcal{MFC}_{\check{\rho},\mathbf{s},\mathbf{t}}^{\mathbf{W},\mathbf{b}}$ is a fully instantiated layer with all of its internal parameters $\mathbf{W}$ and $\mathbf{b}$ defining a function from input to output values.

From Lemma 1 and 2 directly follows:

**Corollary 3.** *For each $i \in \{1, \ldots, n\}$ and each $j \in \{1, \ldots, m\}$ we have:*

*if $t_i = 1$, then $\dfrac{\partial y_j}{\partial x_i} \geq 0$;*      *if $\mathbf{s} = \mathbf{1}$, then $\mathbf{y}_j$ is convex;*

*if $t_i = -1$, then $\dfrac{\partial y_j}{\partial x_i} \leq 0$;*      *if $\mathbf{s} = \mathbf{0}$, then $\mathbf{y}_j$ is concave.*

On the layer level, we can control both monotonicity, convexity and concavity of the output with respect to chosen input variables. The following section discuss how we can use such layers to build practical neural networks with the same properties.

## 3.2 Composing monotonic constrained dense layers

As mentioned before, the main advantage of our proposed monotonic dense unit is its simplicity. We can build deep neural nets with different architectures by plugging in our monotonic dense blocks. The figures Fig 3 and 4 show two examples of neural architectures that can be built using the proposed monotonic dense block.

The first example shown in the figure Fig 3, corresponds to the standard MLP type of neural network architecture used in general, where each of the input features is concatenated to form one single input

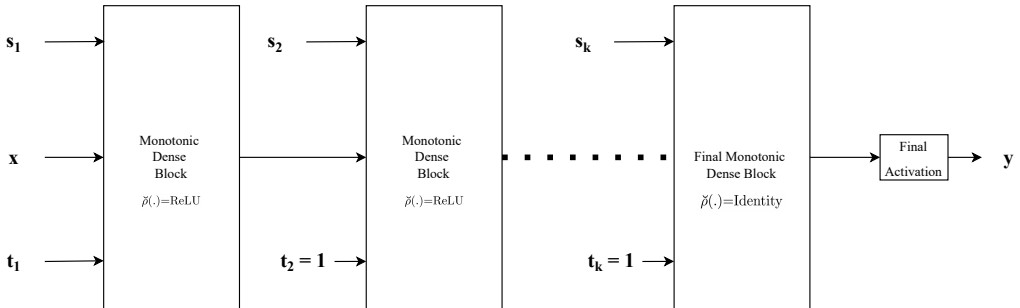

Figure 3: Neural architecture type 1

feature vector $\mathbf{x}$ and fed into the network, with the only difference being that instead of standard fully connected or dense layers, we employ monotonic dense units thorughout. For the first (or input layer) layer, the indicator vector $\mathbf{t}$, is used to identify the monotonicity property of the input feature with respect to the output. Specifically, $\mathbf{t}$ is set to $1$ for those components in the input feature vector that are monotonically increasing and is set to $-1$ for those components that are monotonically decreasing and set to $0$ if the feature is non-monotonic. For the subsequent hidden layers, monotonic dense units with the indicator vector $\mathbf{t}$ always being set to $1$ are used in order to preserve monotonicity. Finally, depending on whether the problem at hand is a regression problem or a classification problem (or even a multi-task problem), an appropriate activation function (such as linear activation or sigmoid or softmax) to obtain the final output.

Figure Fig. 4 shows another example of a neural network architecture that can be built employing proposed monotonic dense blocks. The difference when compared to the architecture described above lies in the way input features are fed into the hidden layers of neural network architecture. Instead of concatenating the features directly, this architecture provides flexibility to employ any form of complex feature extractors for the non-monotonic features and use the extracted feature vectors as inputs. Another difference is that each monotonic input is passed through separate monotonic dense units. This provides an advantage since depending on whether the input is completely concave or convex or both, we can adjust the activation selection vector $\mathbf{s}$ appropriately along with an appropriate value for the indicator vector $\mathbf{t}$. Thus, each of the monotonic input features has a separate monotonic dense layer associated with it. Thus as the major difference to the above-mentioned architecture, we concatenate the feature vectors instead of concatenating the inputs directly. The subsequent parts of the network are similar to the architecture described above wherein for the rest of the hidden monotonic dense units, the indicator vector $\mathbf{t}$ is always set to $1$ to preserve monotonicity.

## 3.3 Universal approximation

The classical Universal Approximation Theorem [7, 17, 28] states that any continuous function on a closed interval can be approximated with a feed-forward neural network with one hidden layer if and only if its activation function is nonpolynomial. In [19], authors prove the approximation property holds for arbitrary *deep* neural networks with bounded number of neurons in each layer holds if the activation function is nonaffine and differential at at least one point.

In [8], authors shows the universal approximation property for constrained multivariate neural networks using *sigmoid* as the activation functions: any multivariate continuous monotone function on a compact subset of $\mathbb{R}^k$ can be approximated with a constrained neural network with the sigmoid activation function of at most $k$ layers (Theorem 3.1). However, the proof of the theorem uses only the fact that the Heavyside function $\mathbf{H}$ defined as

$$H(x) = \begin{cases} 1 & \text{if } x \geq 0 \\ 0 & \text{otherwise} \end{cases}$$

can be approximated with the sigmoid function on a closed interval. It is important to note that this provides an upper bound on the number of monotonic layers in the general case and the number of layers for a particular function should be determined experimentally in practice (in our experiments in Section 4, we got the best performing networks using 2-3 layers).

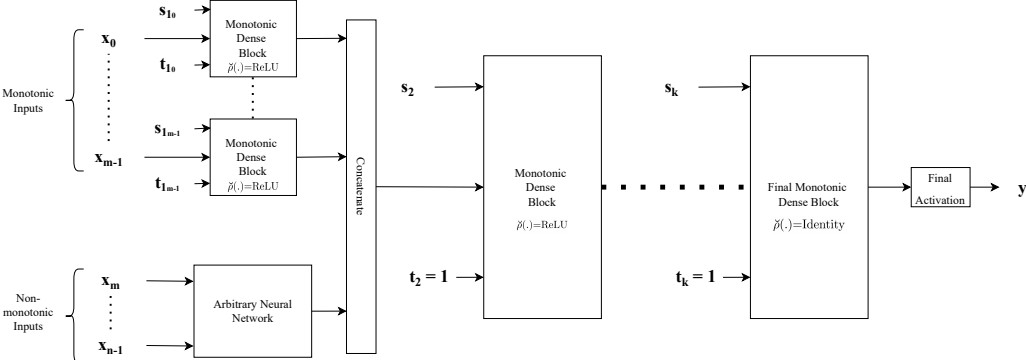

Figure 4: Neural architecture type 2

We provide a simple condition for an arbitrary activation function for the universal approximation property to hold:

**Theorem 4.** *Let $\breve{\rho}$ be a monotone activation function. If a constrained neural network using $\breve{\rho}$ with a single hidden layer can approximate the Heavyside function on a closed interval, then any multivariate continuous monotone function on a compact subset of $\mathbb{R}^k$ can be approximated with a constrained neural network of at most $k$ layers using activation functions $\breve{\rho}$.*

Since we can approximate any continuous monotonic function with monotonic piecewise linear segments, we have:

**Corollary 5.** *A constrained neural network of at most $k$ layers using ReLU as the activation function can approximate any multivariate continuous monotone function on a compact subset of $\mathbb{R}^k$.*

The same property can be shown using in the same way for activation functions other than ReLU.

## 4 Experiments

In order to analyze the practical utility of the proposed method, we experiment with various datasets and compare them with the recent state-of-the-art. For the first set of experiments, we use the datasets employed by authors in [22] and use the exact train and test split for proper comparison. We perform experiments on 3 datasets: COMPAS [18], which is a classification dataset with 13 features of which 4 are monotonic; Blog Feedback Regression [4], which is a regression dataset with 276 features of which 8 are monotonic; Loan Defaulter[1], which is a classification dataset with 28 features of which 5 are monotonic. The dataset contains half a million data points. For comparison with other methods, we compare with *Certified monotonic networks (Certified)* [22] and other methods described in it.

For the second set of experiments, we use 2 datasets: *Auto MPG* (which is a regression dataset with 3 monotonic features) and *Heart Disease* (which is a classification dataset with 2 monotonic features) as employed in the work [39] and once again use the exact train and test split for proper comparison. We compare with the method *COMET* described in [39] along with *Min-Max Net* [8] and *Deep Lattice Network (DLN)* [43] as described in [39].

For the classification tasks, we use cross-entropy and for the regression tasks, we use mean-squared-error as loss functions. We employ gridsearch to find the optimal number of neurons, network depth or layers, batch size, activation function and the number of epochs. For training, we first find the optimal learning rate using learning rate finder [40] and then train with one cycle policy [41].

### 4.1 Results

The results on the dataset above are summarized in Tables 1, 2. It shows that our method tends either match or surpass the other methods in terms of test accuracy for classification tasks and root mean

---

[1]https://www.kaggle.com/wendykan/lending-club-loan-data

squared error (RMSE) for regression tasks. For each of the datasets, we run the experiments ten times after finding the optimal hyperparameters and report the mean and standard deviation of the best five results. Experiment results show that networks learned by our method can achieve better results with fewer parameters, than the best-known algorithms for monotonic neural networks, such as Min-Max Network [8] and Deep Lattice Network [43]. It should be noted that the recent state-of-the-art works-*Certified* [22] and *COMET* [39] require multiple runs in order to even satisfy monotonic constraints whereas monotonicity is guaranteed by simply employing the proposed monotonic dense units.

The most important advantage of our solution is simplicity and computational complexity. Our models have slightly better performance (accuracy or RMSE) on all datasets we tested them on, but it is important to note they have significantly fewer parameters and the simplest training procedure. As such, they reduce the carbon footprint when used in data centers and also aid in the easier adoption of edge computing applications.

| Method | COMPAS [18] | | Blog Feedback [4] | | Loan Defaulter | |
|---|---|---|---|---|---|---|
| | Parameters | Test Acc ↑ | Parameters | RMSE ↓ | Parameters | Test Acc ↑ |
| Isotonic | N.A. | 67.6% | N.A. | 0.203 | N.A. | 62.1% |
| XGBoost [5] | N.A. | 68.5% ± 0.1% | N.A. | 0.176 ± 0.005 | N.A. | 63.7% ± 0.1% |
| Crystal [24] | 25840 | 66.3% ± 0.1% | 15840 | 0.164 ± 0.002 | 16940 | 65.0% ± 0.1% |
| DLN [43] | 31403 | 67.9% ± 0.3% | 27903 | 0.161 ± 0.001 | 29949 | 65.1% ± 0.2% |
| Min-Max Net [8] | 42000 | 67.8% ± 0.1% | 27700 | 0.163 ± 0.001 | 29000 | 64.9% ± 0.1% |
| Non-Neg-DNN | 23112 | 67.3% ± 0.9% | 8492 | 0.168 ± 0.001 | 8502 | 65.1% ± 0.1% |
| Certified [22] | 23112 | 68.8% ± 0.2% | 8492 | 0.158 ± 0.001 | 8502 | 65.2% ± 0.1% |
| **Ours** | **101** | **68.9% ± 0.5%** | **1101** | **0.156 ± 0.001** | **177** | **65.3% ± 0.001%** |

Table 1: Comparison of our method with other methods described in [22]

| Method | Auto MPG | Heart Disease |
|---|---|---|
| | RMSE ↓ | Test Acc ↑ |
| Min-Max Net [8] | 10.14 ± 1.54 | 0.75 ± 0.04 |
| DLN [43] | 13.34 ± 2.42 | 0.86 ± 0.02 |
| COMET [39] | 8.81 ± 1.81 | 0.86 ± 0.03 |
| **Ours** | **3.04 ± 0.13** | **0.86 ± 0.02** |

Table 2: Comparison of our method with other methods described in [39]

# 5   Conclusion

In this paper, we proposed a simple and elegant solution to build constrained monotonic networks which can approximate convex as well as concave functions. Specifically, we introduced a constrained monotone fully connected layer which can be used as a drop-in replacement for a fully connected layer to enforce monotonicity. We then employed our constrained monotone fully connected layer to build neural network models and showed that we can achieve either similar or better results to the recent state-of-the-art ([39, 22] in addition to the well-known works such as Min-Max networks [8] and DLNs [43]. However, the main advantage of the proposed solution is not higher accuracy but its computational and memory complexity: we use orders of magnitude fewer parameters and computation which makes the resulting neural networks more energy efficient.

One drawback of our proposed method is that we are restricted by the choice of activation functions i.e., we can only employ monotonic activation functions. In the future, we would like to build simple monotonic counterparts for other standard neural layers such as convolutional neural networks, recurrent neural networks and their variants, and attention models. Last but not least, we proved such networks can approximate any multivariate monotonic function.

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

## A  Detailed proofs

We restate all lemmas from the main text here are give detailed proofs of them.

The following is well known result, proved here for completeness only:

**Lemma 1.** *For each $i \in \{1, \ldots, n\}$ and $j \in \{1, \ldots, m\}$ we have:*

- *if $t_i = 1$, then $\dfrac{\partial h_j}{\partial x_i} \geq 0$, and*

- *if $t_i = -1$, then $\dfrac{\partial h_j}{\partial x_i} \leq 0$.*

*Proof.* From $\mathbf{h} = |\mathbf{W}|_{\mathbf{t}} \cdot \mathbf{x} + \mathbf{b}$ we get $h_j = \sum_i w'_{i,j} x_i + b_j$. Hence $\dfrac{\partial h_j}{\partial x_i} = w'_{i,j}$. Finally, from equation 2 we have

$$\frac{\partial h_j}{\partial x_i} = \begin{cases} |w_{i,j}| \geq 0 & \text{if } t_i = 1 \\ -|w_{i,j}| \leq 0 & \text{if } t_i = -1 \end{cases}$$

$\square$

**Lemma 2.** *Let $\mathbf{y} = \rho^{\mathbf{s}}(\mathbf{h})$. Then for each $j \in \{1, \ldots, m\}$ we have $\dfrac{\partial y_j}{\partial h_j} \geq 0$. Moreover*

- *if $\mathbf{s} = \mathbf{1}$, then $\rho_j^{\mathbf{s}}$ is convex; and*

- *if $\mathbf{s} = \mathbf{0}$, then $\rho_j^{\mathbf{s}}$ is concave.*

*Proof.* From equation 5

$$\rho^{\mathbf{s}}(\mathbf{h}) = \mathbf{s} \odot \breve{\rho}(\mathbf{h}) + (\mathbf{1} - \mathbf{s}) \odot \hat{\rho}(\mathbf{h})$$

we have:

$$y_j = s_j \breve{\rho}(h_j) + (1 - s_j) \hat{\rho}(h_j)$$

$$\frac{\partial y_j}{\partial h_j} = s_j \breve{\rho}(h_j) + (1 - s_j) \hat{\rho}(h_j)$$

$$= s_j \frac{\partial \breve{\rho}(h_j)}{\partial h_j} + (1 - s_j) \frac{\partial \hat{\rho}(h_j)}{\partial h_j}$$

Since both $\breve{\rho}$ and $\hat{\rho}$ are monotonically increasing and $s_j \in [0, 1]$ we have:

$$\frac{\partial y_j}{\partial h_j} \geq 0$$

if $\mathbf{s} = \mathbf{1}$ or $\mathbf{s} = \mathbf{0}$, we have $\rho^{\mathbf{s}}(\mathbf{h}) = \breve{\rho}(\mathbf{h})$ or $\rho^{\mathbf{s}}(\mathbf{h}) = \hat{\rho}(\mathbf{h})$, which is a convex or a concave function in all components of the output, respectively. $\square$

For completeness, we repeat the Theorem 3.1 from [8] and its proof here:

**Theorem 3.1** *For any continuous monotone nondecreasing function $f : K \to \mathbb{R}$, where $K$ is a compact subset of $\mathbb{R}^k$, there exists a feedforward neural network with at most $k$ hidden layers, positive weights, and output $O$ such that $|O_x - f(x) < \epsilon$, for any $x \in K$ and $\epsilon > 0$.*

*Proof.* The proof is derived by induction on the number of input variables $k$. Without loss of generality, we may assume that $f > 0$ (otherwise, we add a constant $C$ and approximate $f + C$ with the network output $O$, then modify $O$ with a negative bias at the output node). First, we assume that $f$ is strictly increasing and $C^\infty$. In case of $k = 1$, we write

$$f(x) = \int_0^\infty \mathbf{H}\left(f(x) - u\right) du \tag{7}$$

where **H** is the Heavyside function:

$$H(x) = \begin{cases} 1 & \text{if } x \geq 0 \\ 0 & \text{otherwise} \end{cases}$$

Since $f$ is continuous and increasing, it is invertible and therefore the right-hand side of 7 can be written as

$$f(x) = \int_0^\infty \mathbf{H}\left(x - f^{-1}(v)\right) dv \tag{8}$$

The integral can be approximated arbitrarily well by a Riemann sum

$$\sum_{i=1}^N (v_{i+1} - vi)\, \mathbf{H}\left(x - f^{-1}(v_i)\right) \tag{9}$$

where $[v_i]_{i=1}^N$ is a partition of the interval $[f(a), f(b)]$. This expression corresponds to a neural network with input $x$, one hidden layer with $N$ neurons all connected to the input with weight of 1, bias term in the hidden neurons $f^{-1}(v_i)$, and the weights connecting the hidden layer with the output $v_{i+1} - vi > 0$. Note that the Heavyside function **H** can be replaced by a sigmoid activation function using a standard approximation argument.

Assume that Theorem 3.1 holds for $k-1$ input variables. We now combine the integral representation in 7 with the induction assumption. For a given $v$, we may solve the equation of the level set corresponding to to $v$ for $x_k$

$$f(x_1, \ldots, x_k) = v.$$

By the implicit function theorem, there exists a function $g_v$ such that

$$f(x_1, \ldots, g_v(x_1, \ldots, x_{k-1})) = v. \tag{10}$$

Note that $g_v$ is decreasing in all arguments $x_i$. This can be seen by taking the partial derivative of 10 with respect to $x_i$. We will now show that

$$\mathbf{H}\left(f(x) - v\right) = \mathbf{H}\left(x_k - g_v(x_1, \ldots, x_{k-1})\right)$$

analogously to 8 for the 1-D case. Note that it is sufficient to show that

$$f(x) < v \text{ if and only if } x_k < g_v(x_1, \ldots, x_{k-1})$$

and

$$f(x) > v \text{ if and only if } x_k > g_v(x_1, \ldots, x_{k-1}).$$

But this follows easily from 10 and the fact that $f$ is increasing in all its arguments. We now approximate the integral in 7 with a Riemann sum, leading to the following equation analogously to 9:

$$R = \sum_{i=1}^N (v_{i-1} - v_i)\, \mathbf{H}\left(x_k - g_{v_i}(x_1, \ldots, x_{k-1})\right) \tag{11}$$

Since $g_{v_i}$ is decreasing in all its arguments $-g_{v_i}$ is increasing. By the induction assumption, we can approximate $-g_{v_i}$ with a feedforward neural network $O_i$ with $x_1, \ldots, x_{k-1}$ as inputs, $k-1$ hidden layers, and nonnegative weights, such that

$$\left| \sum_{i=1}^N (v_{i-1} - v_i)\, \mathbf{H}\left(x_k - O_i(x_1, \ldots, x_{k-1})\right) - R \right| < \epsilon$$

because the sum is finite. Expression 11 corresponds to a feedforward neural network with $k$ inputs and $k$ hidden layers. Here $k - 1$ hidden layers are needed to represent $-g_{v_i}$ and the $k$-th hidden layer is needed to combine $N$ neural networks with outputs $O_i$ and the input $x_k$. The weights on the connections between the last hidden layer and the final output are $(v_{i+1} - v_i) > 0$. The input $x_k$ is directly (skip-layer) connected to the $k$-th hidden layer.

We can now easily generalize the proof to continuous nondecreasing functions. For continuous functions, we define the convolution of $f$ with a mollifier $K_\delta$ by

$$f_\delta = f \otimes K_\delta$$

Then, $f_\delta$ is $C^{infty}$ and $f_\delta \longrightarrow f$ as $\delta \downarrow 0$ uniform on compact subsets. Furthermore, $f_\delta$ is also increasing since $K_\delta > 0$. Now choose $\delta$ such that $|f - f_\delta| < \frac{\epsilon}{2}$ and approximate $f_\delta$ with a feedforward neural network $O$ such that $|f_\delta - O| < \frac{\epsilon}{2}$. Then, $|f - O| < \epsilon$.

If $f$ is nondecreasing, then approximate $f$ by $f_\delta$

$$f_\delta = f + \delta(x_1 + \cdots + x_k)$$

which is strictly increasing and let $\delta \downarrow 0$.

$\square$

# B    Datasets Description

The descriptions of datasets used for comparison are detailed below. As mentioned in the section 4, the datasets are chosen from [22] and [39] for proper evaluation. The train-test splits of $80\% - 20\%$ are used for all comparison experiments.

1. COMPAS [18] is a binary classification dataset, where the task is to predict risk score of an individual committing crime again two years, based on the criminal records of individuals arrested in Florida. The risk score needs to be monotonically increasing with respect to the following attributes `number of prior adult convictions, number of juvenile felony, number of juvenile misdemeanor`, and `number of other convictions`. It should be noted that there have been ethical concerns with the dataset [18, 32]

2. Blog Feedback [4] is a regression dataset where the task is to predict the number of comments in the upcoming 24 hours from a feature set containing 276 features of which 8 (A51, A52, A53, A54, A56, A57, A58, A59) are monotonic features. The readers are suggested to refer to link [2] for more details. As mentioned by the authors of [22], only the data points with targets smaller than the 90th percentile are used since the outliers could dominate the mean-squared-error metric.

3. Lending club loan data[3] is a classification dataset, where the task is to predict whether the individual would default on loan, from a feature set having 28 features containing data such as the current loan status, latest payment information etc,. The probability of default should be non-decreasing with respect to `number of public record bankruptcies, Debt-to-Income ratio`, and non-increasing with respect to `credit score, length of employment, annual income`.

4. Auto MPG[4] [29] is a regression dataset where the task is to predict city-cycle fuel consumption in miles per gallon (MPG) from a feature set containing 7 features of which the monotonic features are `weight (W), displacement (D)`, and `horse-power (HP)`

5. Heart Disease[5] [11] is a classification dataset, where the task is to predict the presence of heart disease from a feature set containing 13 features of which the risk associated with heart disease needs to be monotonically increasing with respect to the features `trestbps (T), cholestrol (C))`

# C    Additional Experiments and Results Details

For all our experiments, we adopt simple architectures of the types depicted in Figure. 3 or Figure. 4. More often than not, we have seen from our experiments that the neural architecture type 2 (Figure. 4) performs better than the neural architecture type 1 (Figure. 3). The number of hidden layers (apart from the input layer) is either set to be 1 or 2. The number of neurons in the hidden layer is selected from 4, 8, 16, 32, 64. The activation function is set to be either exponential linear unit (ELU) or Rectified Linear Unit (ReLU).

---

[2]https://archive.ics.uci.edu/ml/datasets/BlogFeedback

[3]https://www.kaggle.com/wendykan/lending-club-loan-data

[4]https://archive.ics.uci.edu/ml/datasets/auto+mpg

[5]https://archive.ics.uci.edu/ml/datasets/heart+disease

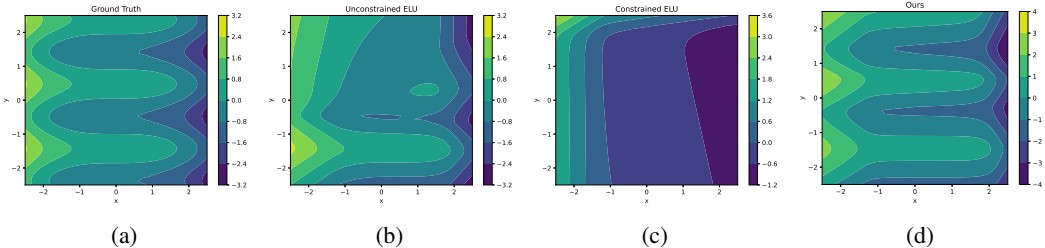

(a)          (b)          (c)          (d)

Figure 5: Results for function fitting experiments where (a) is the ground truth, (b) is the result obtained from a simple unconstrained neural network (with ELU activation) and (c) is the result obtained from a constrained neural network (weights to be positive and ELU activation) and (d) shows our results with constrained weights and bipolar ELU activation

Apart from the experimental results in the main part of the paper, here we highlight the usefulness of the our monotonic dense unit. In a scenario where the dataset is noisy and very small and also contains monotonic features, our neural network constructed using our proposed monotonic dense unit tends to perform better because of the inductive bias. To illustrate this we consider a synthetic dataset generated by function $f(x, y) = sgn(ax) \ x^3 + b \ \sin(cy), a, b, c \in \{0.5, 0.35, 3.3\}$. We sample points uniformly in the range $(-2.5, 2.5)$ for both $x$ and $y$, and add zero-centered Gaussian noise. We then sample only 100 points for training and train three neural networks having the same kind of architecture but having difference in constraints on weights as mentioned in section 3, but activation function used is ELU instead of ReLU. We test the three networks - Unconstrained, Constrained ELU and Constrained bipolar ELU, to on the task of fitting the aforementioned function. The results be seen in the Figure. 5. It is evident that the unconstrained neural network does not preserve monotonicity and although constrained neural network does preserve monotonicity, it cannot faithfully reproduce the concave part of the function, whereas ours (constrained bipolar ELU) can fit both concave and convex part of the function.

