# OpenReview forum: "Constrained Monotonic Neural Networks"
_NeurIPS.cc/2022/Conference — NeurIPS 2022 Submitted_

### Official Review · Reviewer_8Y4o · 2022-07-10

**Rating:** 6
**Confidence:** 3
**Soundness:** 3 good
**Presentation:** 3 good
**Contribution:** 3 good

**Summary:**

The authors proposed a constrained monotonic dense layer architecture that is compositable to form a monotonic neural network which can approximate non-convex function and has strong empirical results.

**Questions:**


1. On the limitation of the stacking multiple layers, would it be possible to adapt this method to build monotonic layers other than dense layers (such as self-attention or convolutional layers)?

2. In the experiment section, grid search is performed to find the best hyper-parameters for the current method. Would it be possible to perform a similar search for one of the other methods in Table 1 to verify the advantage of the new architecture?

**Strengths And Weaknesses:**

Strengths:

1. The proposed monotonic dense layer employs simple, off-shelf components in today's deep learning frameworks, making it very easy to use;
2. The design of the monotonic layer allows composition, making it extensible to deep neural networks;
3. Strong empirical results (models with fewer parameters and better quality).


Weakness:

1. Several definitions are missing from the writing: in section 3.1., definition of a partially monotonic function and a compact segment should be given for clarity.

2. If I understand correctly, in the composed monotonic model with both monotonic and non-monotonic inputs, the layers after the concatenation layer can only be monotonic dense layers. This seems to pose a limitation on stacking multiple layers, such as a stack of transformer blocks, or a stack of convolutional layers.

---

> ### Author Response · Authors · 2022-08-02
> **Response**
>
> Thank you very much for the review. Regarding the concerns and questions please find our responses inline below:
>
>
> “Several definitions are missing from the writing: in section 3.1., definition of a partially monotonic function and a compact segment should be given for clarity.”
>
> Response: Thank you for pointing it out, appropriate modifications have been made in the paper.
>
>
> “If I understand correctly, in the composed monotonic model with both monotonic and non-monotonic inputs, the layers after the concatenation layer can only be monotonic dense layers. This seems to pose a limitation on stacking multiple layers, such as a stack of transformer blocks, or a stack of convolutional layers.”
>
> Response:
> - Transformer blocks or stacks of convolutional layers should be used as depicted in Figure 4 in the place labed arbitrary neural network, taking only non-monotonic parameters as inputs. The idea behind this architecture is to use complex modern neural networks on non-monotonic parameters and mix them with the monotonic ones in the final layers of the network. The monotonic functions are a bit unexciting, pun intended, since they are rather smooth and always increasing or decreasing. There is no need for a complex machinery to model them, the complexity is typically contained in the non-monotonic parameters.
>
> “On the limitation of the stacking multiple layers, would it be possible to adapt this method to build monotonic layers other than dense layers (such as self-attention or convolutional layers)?”
>
> Response:
> - We can extend the convolutional layers to be partially monotonic with respect to some of the features, but that’s not was is typically needed in the real world applications. We find that the architecture in Figure 4 is more suited for most of the problems where complex neural networks operate on non-monotonic features and get mixed with the monotonic ones in the final layers of the network.
>
>
> “In the experiment section, grid search is performed to find the best hyper-parameters for the current method. Would it be possible to perform a similar search for one of the other methods in Table 1 to verify the advantage of the new architecture?”
>
> Response:
> - We compare with the best performing methods as mentioned in the reference numbers [20] and [36]. We employ the same datasets with the exact train and test split used by the authors. According to reference [36], they too tabulate the results for best perming networks after grid search, and the reference [20] also report the best performing results in their main table after experimenting with different hyperparameters. Hence we can say that the comparison results reported in our paper too are based on search done by the authors of respective papers.

---

### Official Review · Reviewer_ShEg · 2022-07-11

**Rating:** 3
**Confidence:** 4
**Ethics Flag:** Yes
**Soundness:** 1 poor
**Presentation:** 1 poor
**Contribution:** 2 fair

**Summary:**

In the current paper, the authors propose a deep neural network that produces monotonic functions. It is easy to produce a convex function (constrain the weight to non-negative values) but it is not trivial to get monotonic function.   To that end, the author introduce a so called Monotonic Dense Unit that combine convex and concave activation functions in order to output a provably monotonic function.                                                                   The authors then justify their approach on 3 benchmarks.

**Questions:**

How do you compute the vector t (monotonicity indicator) in practice?

What is the advantage of your method over constrained networks? Do you have examples that highlights the benefits of Monotonic Dense Unit?

As mentioned, the question of universality is particularly interesting, could you describe the reach of your algorithm in term of descriptive power?

What would prevent to use any other concave decreasing activation function instead of the point symmetric?

**Ethics Review Area:**

["Inadequate Data and Algorithm Evaluation"]

**Limitations:**

The author must discuss their usage of COMPAS dataset, as well as the comment l29-l33 as already stated in the main review.

**Strengths And Weaknesses:**

# Strengths

Modeling functions with specific constraints such as monotonicity or convexity is an interesting and broad topic of ML with many application (despite the thoughts of the authors l139). The proposed method is sounds and provably outputs monotonic function in selected components.

# Weaknesses

The paper lacks proper definition about what are increasing and decreasing features and how do they interact with each other.                                                                                                               The propose method is very incremental and the paper lacks of a proper develop theory. The author conjecture l189 the universality of their approach but to not provide elements. Also, the PMU seems a bit over-complex and tend to look like an engineer approach.
                                                                                                                                                                                                                                                                                                                                                                                               * Global style issues

The paper is written in a somehow bad style. Nothing is properly defined (the core of the paper is "defined" in the caption of figure 2), the text is full of pleonasms such as "monotonic increasing functions" and often contradicts one paragraph with another (e.g. l36 claims it is not trivial, l49 asserts it is).                                                                                                                                                               A lot of points are repeated over and over, sometimes with useless information (e.g. the name of bozillions activations function from the literature) and it often feels like the author just wanted to fill 9 pages. This gives room to properly introduce what is the code of the paper and all the needed notions.                                                                                                                                                                   Whether an equation is a definition or a proposition is often unclear (e.g. l237 seems to be a proposition (because of 'then') but is in fact the definition).                                                                              Two names are proposed for the global algorithm "Monotonic Dense Unit" or "Constrained Monotone Fully Connected Layer" (figure 1, l.219-220) none of those being really used in the later. Definition 3 is yet another "Monotone Constrained Fully Connected Layer".

* Ethics

And seriously what the actual heck is the following (l29-33):                                                                                                                                                                               "All other things being equal, a larger house should be deemed more valuable, a bank's client with higher income should be eligible for a larger loan, an offender with a longer crime history should be predicted as more likely to commit another crime, etc. A model without such a monotonic property would not, and certainly should not, be trusted by the society to provide a basis for such important decisions."                                                              This has NOTHING to do in an ML paper.                                                                                                                                                                                                                                                                                                                                                                                                                                                  The use of COMPAS dataset should be ethically discussed.

* More precisely among much more

How to "constrain" the weights is never properly said (only in the introduction, I guess the author mean that the weights are >=0 elsewhere too).

Definition 1: h is defined twice in an inconsistent manner (let h be [...], h=...).

Section 2.4: the question of universality is indeed interesting but this section is somehow very weak if the authors have nothing to say on this topic.

Figure 2: why are W "*kernel* weight"

I do not understand Figure 4 and the explanation (l280-288) is really confusing.
                                                                                                                                                                                                                                                                                                                                                                                                   Eq (1): the case = 0 is ill-defined

* Global remark

A major limitation of the proposed approach is that the assumed monotonic components of the input are dealt in an independent manner.

* Experiments:
    - the experimental protocol is insufficiently described;
    - the networks used are never properly described;
    - dataset are not described well enough, and the hypothesis (monotonicity) never discussed;
    - not enough experiments, I would also have expected some ablation studies and hand-crafted problems;
    - the use of COMPAS dataset must trigger some ethical concerns that are ignored in the paper;
    - the choice of datasets and the chosen "monotonic" target features should be accurately discussed;
    - the proposed model should at least be compared with "constrained NN" in order to assert the improvement of the proposed algorithm over naivety.

* Typos

Punctuation is missing in many if not most equations.

References: many if not most references are incomplete (arxiv instead of peer reviewed journal/conferences)

---

> ### Author Response · Authors · 2022-08-02
> **Response**
>
> Thanks for the review. Please find our responses inline.
>
>
> “How to "constrain" the weights is never properly said (only in the introduction, I guess the author mean that the weights are >=0 elsewhere too).”
> - The constraining operation is indeed described in Equation 2.
>
> “Definition 1: h is defined twice in an inconsistent manner (let h be [...], h=...).”
> - Thanks for pointing it out,  we have corrected it.
>
>
> “Section 2.4: the question of universality is indeed interesting but this section is somehow very weak if the authors have nothing to say on this topic.”
> - We felt that was the main challenge with the paper and in the revised version of the paper we proved the universal approximation property for the ReLU activation for the class of continuous monotonic functions on a compact subset.
>
>
> “Figure 2: why are W "kernel weight"”
> - It is relative common to employ the term “kernel weights” when describing a single dense layer unit since -
>
> output = activation(dot(input, kernel) + bias)
>
> “I do not understand Figure 4 and the explanation (l280-288) is really confusing.”
> - Thank you for pointing that out. There was a version mismatch between the text and the figure that was corrected in the revised version of the paper.
>
> Concerns regarding Experiments -
>
> - We used the same experimental protocol, the same datasets, tha same train/test splits and the same evaluation code from  Liu et. al  and Sivaraman et. al. (References [29] and [36] in the initial submission of paper) to make direct comparison as easy as possible. We included descriptions of the datasets in the appendix of the revised paper to make it self contained.
>
> - To make comparison with the existing solutions as unbiased as possible, we used their datasets. The main advantage of our solution is simplicity in the sense of computational and memory requirements and our goal was to show the proposed solution is as good as the alternatives while requiring musg less resources.
>
> the use of COMPAS dataset must trigger some ethical concerns that are ignored in the paper;
> - Regarding the COMPAS dataset, we used the dataset for comparison from the repository provided by Liu et. al  [29]. Moreover, we have cited an apt paper regarding the ethical concerns of COMPAS dataset as early as in the introduction section itself: Cynthia Rudin, Caroline Wang, and Beau Coker. The age of secrecy and unfairness in recidivism prediction.Harvard Data Science Review, 2(1), 3 2020
>
> the proposed model should at least be compared with "constrained NN" in order to assert the improvement of the proposed algorithm over naivety.
>
> Response: We have indeed compared with constrained NN, the method Non-Neg-DNN in Table 1 corresponds to the “naive” version of the constrained NN.
>
> “How do you compute the vector t (monotonicity indicator) in practice?”
>
> Response:
> - The intended use of the monotonicity vector t is to be specified as the input of the network. The specification could be the result of legal, ethical and/or safety concerns. Alternatively, it can be the result of statistical tests as cited in  Liu et. al (References [29])
>
>
> “What is the advantage of your method over constrained networks? Do you have examples that highlights the benefits of Monotonic Dense Unit?”
>
> Response:
> - Our method allows for use of unsaturated monotonic activation functions such as ReLU and ELU in approximating non-convex monotonic neural networks. The naive approach allowed only for saturated activation functions such as sigmoid to be used, which are more difficult to train and have inferior performance in terms of accuracy or RMSE.
>
> “As mentioned, the question of universality is particularly interesting, could you describe the reach of your algorithm in term of descriptive power?”
>
> Response:
> - You are absolutely right and that was our main focus since the initial submission of the paper. In the revised version of the paper, we proved the universal approximation property for the ReLU activation function and provided a sufficient condition for other activation functions for the property to hold. Please see Section 3.3 and Appendix for the details.
>
>
> “What would prevent to use any other concave decreasing activation function instead of the point symmetric?”
>
> Response:
> - Yes, indeed any concave decreasing activation function can be used, the only restriction being that activation function is monotonic. In fact, we did an experiment with two different activation functions, but in practice we found point reflection to be more stable than using two different activation functions. We also experimented with making vector s learnable, but it only made training more difficult while results were not statistically better. The simplest solution was as least as good as any other more complicated construction we tested.

---

> > ### Comment · Reviewer_ShEg · 2022-08-05
> > **Universality proof does not hold.**
> >
> > Thank you very much for the answer to my questions. I will need a little bit more time to digest the new version of the paper, as well as going over the answers to the other reviewers.
> > However I took some time to look at the result of universality that has been annouced on the rebuttal, and its proof (theorem 3.1 in the supplementary material).
> > I have many remarks and question about it.
> > This proof is caracteristic of many problems I pointed all over the paper so I'll develop a little bit here.
> >
> > Statement of the theorem:
> > - (i) As in the paper (at least its first version), the notion of monotonicity is not defined (it is obvious for a single variable, but what about multiple-dimension?). So I assume the theorem should be read "Any continuous function $f : K \subset \mathbb{R}^d \rightarrow \mathbb{R}$ with non-decreasing components (this should be of course written in a more rigorous way in the paper).
> > - (ii) In conclusion of the theorem (the universality result) a number $k$ of hidden layers is given but this $k$ is neither defined, neither constrained, neither interact with anything else.
> > - (iii) The theorem should be "for all epsilon, there exist foobars(epsilon)" and not "there exist foobars, for all epsilon(foobars)" which is different.
> >
> > Proof of the theorem:
> > - The Riemann sum argument with the Heavyside function is bizarre and I don't really see where it could go, but why not;
> > - assumptions are needed in order for the sum described at Eq. (9) to converge;
> > - what is the "level v" and how does it relates with the previous Riemann sum argument? Do we need f to be surjective?
> > - I guess line 524 you consider $f: \mathbb{R}^{k+1} \rightarrow \mathbb{R}$?
> > - Implicit function theorem assumptions should be verified before application. Even though it is somehow easy here I think thank to the increasing in each component assumption, it is not trivial and should be stated.
> > - Implicit function theorem does give the existence of $g_v$ in a **neighborhood** of $x$.
> > - I do not find it is trivial to get the sign of the derivatives of $g_v$, I think the author should give a complete argument.
> >
> > But most importantly I think the proof does not hold:
> > 1) Implicit function theorem is local, as such I do not believe that the equation after Eq (10) holds for all x (as it should), neither the inequalities, and neither the following. Or it should be proven.
> > 2) $\epsilon$ in the equation following Eq (11) is not the $\epsilon$ of the induction hypothesis. It makes me wonder if my comment of the theorem (iii) is a typo or an actual mistake.

---

> > > ### Author Response · Authors · 2022-08-06
> > > **Misunderstanding about cited work**
> > >
> > > Thank you for your comments, but we believe there was potentially a misunderstanding here. As stated in the line above the Theorem 3.1 and its proof in Appendix:
> > >
> > > "For completeness, we repeat the Theorem 3.1 from [8] and its proof here:”,
> > >
> > > the Theorem 3.1 is stated and proved in the cited paper:
> > >
> > > [8] Hennie Daniels and Marina Velikova. Monotone and partially monotone neural networks. IEEE Transactions on Neural Networks, 21(6):906–917, 2010.
> > >
> > > This is a well known and widely cited paper proving the universal approximation property for constrained neural networks with sigmoid activation function. We use it to prove the universal approximation property of our construction using monotonic non-saturated activation functions and its point reflection. We are not aware of any errors found in [8] and we did not find any, although there is always a possibility of that.

---

> > > > ### Comment · Reviewer_ShEg · 2022-08-09
> > > > **error about cited work**
> > > >
> > > > Indeed I am sorry, I missed the fact it is the reproduction of a proof in this 2010 paper. It seems I have a lot to say about it (even though I believe the theorem must be true) ;)
> > > > I appreciate the update of the paper that has been published, I however think that many presentation points are still problematic in my opinion.
> > > > - The ethical concerns have not really been addressed, probably a good way of doing so would be to add many benchmark, and change the phrasing about the use of the COMPAS dataset. For example the usage of "risk score", or the no discussion about the many bias that are present in this dataset make it a very poor benchmark.
> > > > - The sentence about society that should give loans based on income, or sentence crime based on crime history etc is still present in the introduction.
> > > > - Even though the author provided amazing efforts at making the paper much clearer and correcting some typos (many are still there), I have difficulties to go beyond the fuzziness of the paper. By this I mean that many choices are taken and we do not really know why, the vocabulary and explanations are often imprecise or sound weird (like calling a d-dimensional compact a 'segment', or calling 'parameter' what should be called a component).
> > > > From the other reviewers comments it seems to be a personal taste issue, and I will not try to push it against this work. But I strongly believe that mathematical writing should have much higher standards than what is presented in this paper.

---

> > > > > ### Author Response · Authors · 2022-08-10
> > > > > **Response**
> > > > >
> > > > > Thank you for your comments. Please find our responses below.
> > > > >
> > > > > “Indeed I am sorry, I missed the fact it is the reproduction of a proof in this 2010 paper. It seems I have a lot to say about it (even though I believe the theorem must be true) ;)”
> > > > >
> > > > > Response:
> > > > > Thank you for acknowledging this.
> > > > >
> > > > >
> > > > > “The ethical concerns have not really been addressed, probably a good way of doing so would be to add many benchmark, and change the phrasing about the use of the COMPAS dataset. For example the usage of "risk score", or the no discussion about the many bias that are present in this dataset make it a very poor benchmark.”
> > > > >
> > > > > Response:
> > > > > Monotonicity is a well known problem and there are a number of alternative solutions, the most prominent being Deep Lattice Networks by You et. al [43] and Certified Monotonic Neural Networks by Liu et. al. [22]. We found a computationally less expensive solution and we wanted to make a fair comparison with previous work in terms of its performance. We could have picked and chosen datasets and benchmark protocols where our solution performs the best, but we chose to use the exact datasets and benchmark protocols used in the above mentioned works instead. We believe this is the most direct and fair way to compare our work with the previous ones. It turns out that the solution presented here either matches or outperforms all alternative solutions on all datasets their respective authors selected.
> > > > >
> > > > > COMPAS dataset is one of the datasets used in [22] and using our direct comparison methodology, we included it in our benchmarks. The ethical concern about the use of the dataset was addressed in the same way as in the previous works, by citing the following paper where ethical implications of the COMPAS dataset are discussed in length:
> > > > >
> > > > > [32] Cynthia Rudin, Caroline Wang, and Beau Coker. The age of secrecy and unfairness in recidivism prediction. Harvard Data Science Review, 2(1), 3 2020.
> > > > >
> > > > > and by adding the following sentence to the paper:
> > > > >
> > > > > Due to legal, ethical and/or safety concerns, monotonicity of predictive models with respect to some input or all the inputs is required in numerous domains such as financial (house pricing, credit scoring, insurance risk), healthcare (medical diagnosis, patient medication) and legal (criminal sentencing) to list just a few. All other things being equal, a larger house should be deemed more valuable, a bank’s clients with higher income should be eligible for a larger loan [36], and an offender with a longer crime history should be predicted as more likely to commit another crime [32], etc.
> > > > >
> > > > >
> > > > >
> > > > > “The sentence about society that should give loans based on income, or sentence crime based on crime history etc is still present in the introduction.”
> > > > >
> > > > > Response:
> > > > > The sentence in question is:
> > > > >
> > > > > “All other things being equal, a larger house should be deemed more valuable, a bank’s clients with higher income should be eligible for a larger loan [36], and an offender with a longer crime history should be predicted as more likely to commit another crime [32], etc.”
> > > > >
> > > > > These are common requirements on prediction models and similar statements are made in previously accepted NeurIPS papers such as :
> > > > >
> > > > > - in Liu et. al [22]: “For real-world scenarios with fairness or security concerns, model predictions that violate monotonicity could be considered unacceptable. For example, when using ML to predict admission decisions, it may seem unfair to select student X over student Y, if Y has a higher score than X, while all other aspects of the two are identical.”
> > > > >
> > > > > - Sivaraman et. al. [36]: “ For example, one would expect an individual with a higher salary to have a higher loan amount approved, all else being equal, and the speed of a drone to decrease with its proximity to the ground.”
> > > > >
> > > > >
> > > > >
> > > > > “Even though the author provided amazing efforts at making the paper much clearer and correcting some typos (many are still there), I have difficulties to go beyond the fuzziness of the paper. By this I mean that many choices are taken and we do not really know why, the vocabulary and explanations are often imprecise or sound weird (like calling a d-dimensional compact a 'segment', or calling 'parameter' what should be called a component). From the other reviewers comments it seems to be a personal taste issue, and I will not try to push it against this work. But I strongly believe that mathematical writing should have much higher standards than what is presented in this paper.”
> > > > >
> > > > > Response:
> > > > >
> > > > > Thank you again for pointing out the above mentioned points, it helped us make the paper much clearer. We found a few more typos in the meantime, but nothing substantial.

---

### Official Review · Reviewer_k32L · 2022-07-13

**Rating:** 7
**Confidence:** 4
**Soundness:** 4 excellent
**Presentation:** 4 excellent
**Contribution:** 4 excellent

**Summary:**

The paper proposes a differentiable neural network layer which is monotonic in its input by construction. It is defined as a convex combination of a convex and a concave monotonic functions on the input. The layer is simple, both conceptually and to implement, and may be useful to reduce parameter count when the modeler has prior knowledge that the function to be learned is monotonic. This layer is also computationally efficient: it has the same amount of computation as a typical feed forward layer followed by a nonlinearity, up to a constant factor.
After describing the layer, the authors provide empirical validation on some tasks, showing that models using the proposed layer achieve better accuracy with smaller parameter count vs monotonic layers in the literature.

**Questions:**

- The authors discuss convex monotonic networks [1]. On the other hand, they do not discuss Input convex neural networks [2]. Taking the derivative of an input convex neural network wrt input would also yield an nondecreasing function. How would this compare to the proposed model?

[1] Xingchao Liu, Xing Han, Na Zhang, and Qiang Liu. Certified monotonic neural networks. NeurIPS 2020

[2] Amos, B., Xu, L., & Kolter, J.Z. (2017). Input Convex Neural Networks. ArXiv, abs/1609.07152.


**Limitations:**

The paper proposes a class of parametric functions to approximate monotonic functions. It does not provide a guarantee that this class can approximate any monotonic function.

**Strengths And Weaknesses:**

Strengths:
Constraining a model to be input monotonic is an important problem in ML. Monotonicity can stem from either domain knowledge (price of a house should be increasing wrt surface all else fixed), or functional constraints (estimating a cdf).
The proposed method is simple, which is a strength. Although the specific shape of the layer seems to have been explored by previous works, it has not been used in the specific context of monotonicity.
The paper is well written and clear, although perhaps more space could be used for empirical results, and discussing settings in ML where monotonicity is important.
The context of the work is clear.

Main criticism:
The paper does not provide an approximation theoretical result. Are there increasing functions that cannot be approximated by the proposed method?

---

> ### Author Response · Authors · 2022-08-02
> **Response**
>
> Thank you very much for your review and comments. Also, thank you for the rating,  we have addressed your main criticism. Please find our responses inline.
>
>
> Main criticism: “The paper does not provide an approximation theoretical result. Are there increasing functions that cannot be approximated by the proposed method?”
>
> Response:
> - You are absolutely right and that was our main focus since the initial submission of the paper. In the revised version of the paper, we proved the universal approximation property for the ReLU activation function and provided a sufficient condition for other activation functions for the property to hold. Please see Section 3.3 and Appendix for the details.
>
>
> Questions:
> “The authors discuss convex monotonic networks [1]. On the other hand, they do not discuss Input convex neural networks [2]. Taking the derivative of an input convex neural network wrt input would also yield an nondecreasing function. How would this compare to the proposed model?”
>
> Response:
> - Thanks for the comments, we overlooked that paper initially. This is certainly a possible, but more complex way to construct a monotonic model. In the revised version of the paper, we added the following paragraph in the related work section:
> “Given a model with a convex output function, it is possible to use backpropagation [34] to make a monotonic model by computing the derivation of the output function. One simple way to construct a convex function is to use an unsaturated monotonic activation function in a fully connected layer as mentioned above, but we could also use more elaborate architectures such as the input convex neural networks [ 1]. Although possible, these kinds of constructions are computationally more complex than a simple solution proposed here.”

---

### Official Review · Reviewer_QsCV · 2022-07-14

**Rating:** 4
**Confidence:** 4
**Soundness:** 2 fair
**Presentation:** 3 good
**Contribution:** 3 good

**Summary:**

The authors propose a simple scheme based on the convex combination of a function and its point reflection in order to produce a monotonic building block from which input monotonic (or partially input monotonic) neural networks can be built.  The construction is validated both theoretically and experimentally.

**Questions:**

See strengths + weaknesses.

**Limitations:**

Seem to be adequately addressed.

**Strengths And Weaknesses:**

Originality:  There is quite a bit of work on monotonicity in neural networks, and at least to my knowledge, the authors seem to do a good job of positioning their work.  While point reflections have been used in the past, their application here does appear to be novel.

Quality:  I didn't find any serious errors with the technical content.  However, I think that the experiments fall a bit on the short end:  the number of data sets is limited and they are somewhat smallish and the performance improvements in many cases are either not statistically significant or too small to get really excited about.  As a result it feels like a more robust set of experiments, including on problems where monotonicity isn't even required, would significantly bolster the case for acceptance.  Additionally a careful critique of the competing methods in the experiments should be done to emphasize the strengths and weaknesses of the present approach.

Clarity:  I found the article easy to read for the most part.  Some of the definitions could be written more clearly with less fluff, e.g., in Definition 1 use R^n, etc. to simplify the presentation, and there are a few minor typos here and there, e.g., line 268.  I also feel like way too much of the main text is devoted to background work (some of which is repeated way too much -- I got the point the first time).  I'd prefer more experiments to supplant it.

Significance:  The construction is simple and the resulting networks are seemingly easy to train.  As a result, I could see this work being of practical significance.

---

> ### Author Response · Authors · 2022-08-02
> **Response**
>
> Thank you very much for your review and comments. Please find our responses inline.
>
> “ I didn't find any serious errors with the technical content. However, I think that the experiments fall a bit on the short end: the number of data sets is limited and they are somewhat smallish and the performance improvements in many cases are either not statistically significant or too small to get really excited about."
>
> Response:
> - The datasets for experiments  were chosen in such a way that direct comparison with existing methods is possible, Hence we employ the datasets used in the papers Liu et. al  and Sivaraman et. al. (References [29] and [36] in the initial submission of paper). The largest dataset used had the size of ~400K samples. We did not search for larger datasets because as the dataset size grows, the differences in results get much smaller and there is not much improvement in metrics. The difference in accuracy is due to inductive bias in the model and even non-monotonic models converge into monotonic with enough data (although there is no guarantee for any particular dataset size).
>
>
>  "As a result it feels like a more robust set of experiments, including on problems where monotonicity isn't even required, would significantly bolster the case for acceptance. "
>
> Response:
>
> - This is a valid concern, but we feel it is outside the scope of the paper. We feel monotonicity is an important requirement and our focus was on a simple, computationally inexpensive practical solution for an existing problem. Including inductive biases such as monotonicity will increase accuracy of the models even when the monotonicity is not required and we referred readers to Peter Velickovic’s PhD thesis on the subject.
> - Nonetheless, in a scenario where the dataset is noisy and very small and also contains monotonic features, neural networks constructed using our proposed monotonic dense unit tends to perform better because of the inductive bias. We have added an experiment in Appendix Section C  to illustrate this.
>
>
> "Additionally a careful critique of the competing methods in the experiments should be done to emphasize the strengths and weaknesses of the present approach.”
>
> Response:
> - Thank you for pointing that out. We added the following clarification to the Results section:
> The most important advantage of our solution is simplicity and computational complexity. Our models have slightly better performance (accuracy or RMSE) on all datasets we tested it on, but it is important to note they have significantly less parameters and the simplest training procedure. As such, they reduce carbon footprint when used in datacenters and also aid in easier adoption of edge computing applications.
>
>
> “Clarity: I found the article easy to read for the most part. Some of the definitions could be written more clearly with less fluff, e.g., in Definition 1 use R^n, etc. to simplify the presentation, and there are a few minor typos here and there, e.g., line 268.”
>
> Response: Thanks for the suggestion, it has been rewritten in the revised version.
>
>
> “I also feel like way too much of the main text is devoted to background work (some of which is repeated way too much -- I got the point the first time). I'd prefer more experiments to supplant it.”
>
> Response:
> - Thank you for pointing out, we significantly reduced the background section. In addition to the existing comparison results with two recent state-of-the-art methods, we included synthetic experiment results in the Appendix section C. We also used the additional space to include the proof of the universal approximation property. We showed our solution can approximate arbitrary continuous monotonic function on a compact subset of R^n.

---

> > ### Comment · Reviewer_QsCV · 2022-08-08
> > **Thanks for your feedback**
> >
> > Your clarifications are useful, but could you comment on the statistical significance of the experimental results?

---

> > > ### Author Response · Authors · 2022-08-08
> > > **Response**
> > >
> > > We performed experiments on both the classification and regression tasks. Although we compare with all the recent state-of-the-art, our main competing method is Deep Lattice Networks (DLN), since both are monotonic architectures by construction. Compared to DLN, we find that on regression tasks, we obtain statistically significant better performance in-terms of RMSE:
> > >
> > > - **3.04** ± 0.13 vs. **13.34** ± 2.42 on the Auto MPG dataset and
> > > - **0.156** ± 0.001 vs. **0.161** ± 0.001 on the Blog Feedback dataset.
> > >
> > > On the classification tasks, the difference in accuracy on the tested datasets is either slightly better or matching:
> > >
> > > - **68.9%** ± 0.5% vs. **67.9%** ± 0.3% on the COMPAS dataset and
> > > - Matching **86% ± 2%** on the Heart dataset
> > >
> > > The difference in accuracy between competing models decreases with dataset size as inductive bias introduced by monotonicity constraint loses its significance. But the main point  to note is the simplicity in model construction and training when compared to DLNs. The number of training parameters is significantly lesser in our method (by two orders of magnitude when compared to DLNs).
> > >
> > > When compared to the Certified monotonic models, the best performing approach so far, our method performs slightly better or is matching its performance on all datasets. Apart from our models using significantly less parameters (an order or two magnitude less), the certified monotonic networks are even more time consuming to train as they require multiple training/certification cycles to obtain a monotonic network (if certification fails, we need to train a model again until the certification succeeds). Again, we feel the simple and computationally significantly less expensive solution has its advantages in many settings.

---

### Review · Ethics_Reviewer_L7WK · 2022-08-06

**Recommendation:**

The changes below could be implemented in the current version of the paper:

- the quoted sentence from the introduction should be removed and replaced with specific, evidence backed statements about why monotonicity is a goal without making value judgements and stating them as fact
- the checklist answers could be updated as described aabove
- how monotnic vs not features are determined could be added

**Ethical Issues:**

Yes

**Ethics Review:**

The introduction to the paper makes a socially-charged statement as if it were established fact and cites papers within that sentence that do not support the claim of the sentence.

> All other things being equal, a larger
> house should be deemed more valuable, a bank’s clients with higher income should be eligible for a
> larger loan [36], and an offender with a longer crime history should be predicted as more likely to
> commit another crime [32], etc.

this should be treated with greater care.  While monotonicity may be a goal, it is *a* goal, not the only goal and the idea that this is a universal truth represents an oversimplification of the world.  For example, a Black person who lives in an overpoliced community with a longer arrest record may not deserve to be predicted as more likely to commit another crime as compared to a white person who lives in a minimally patrolled community.

How "monotonic features" are determined in each dataset is not described and it should be.  Developing a problem formulation like this, is, ultimately a normative, not neutral activity and each of these choices should be declared and described.  The authors are presumably making these judgements themselves and this should be declared.


The authors said [N/A] to checklist questions 4d-e. Data from people was used, so these questions *do* apply to the paper.  The COMPAS dataset as released by Propublica in fact includes personally identifable information.  It was collected via public records requests, but these individuals were arrested, they never gave consent for the ML use of their data. The paper is of course not alone in this, but the matter does apply to the paper. Information about all datatsets at this level should be included and properly acknowledged in the checklist.

---

### Review · Ethics_Reviewer_7Zwg · 2022-08-09

**Recommendation:**

Given the limitations of the COMPAS dataset, I think the authors could just use another dataset with the monotonicity constraint property. There are a lot of datasets out there with this property. But the authors should discuss the issues with the COMPAS data in greater detail if they want to keep the results in there.

**Ethical Issues:**

Yes

**Ethics Review:**

The authors have stated that they are aware that at least one of the datasets they have used (COMPAS) has ethical issues. Nevertheless, the authors don't really discuss what the ethical issues are. Several other researchers have pointed out limitations with the dataset, including measurement error, bias in the data generation process, and errors in data processing:
https://datasets-benchmarks-proceedings.neurips.cc/paper/2021/file/92cc227532d17e56e07902b254dfad10-Paper-round1.pdf
https://arxiv.org/abs/1906.04711

---

### Author Response · Authors · 2022-08-05
**Paper has been revised significantly**

Dear Reviewers,

Thank you for taking the time to review our work.
- We have made significant changes to the paper, including the proof for universal approximation property for the ReLU activation function
- We have responded to each reviewer individually and uploaded a revised copy of the paper incorporating the feedback we received.

---

### Meta-Review · Area_Chair_yPBa · 2022-08-25

**Recommendation:** Reject
**Confidence:** Less certain

**Metareview:**

There are some ethical issues about using COMPAS dataset in the paper. The ethics reviewers recommend that
1) Given the limitations of the COMPAS dataset, the authors could just use another dataset with the monotonicity constraint property. There are a lot of datasets out there with this property. But the authors should discuss the issues with the COMPAS data in greater detail if they want to keep the results in there.
2) The changes below could be implemented in the current version of the paper:
- The quoted sentence from the introduction should be removed and replaced with specific, evidence backed statements about why monotonicity is a goal without making value judgements and stating them as fact.
- The checklist answers could be updated as described above.
- How monotonic vs not features are determined could be added.

The reviewer also suggests the authors do more benchmark and revise the mathematical writing since it makes the readers hard to follow. Although the paper is in borderline, most of the reviewers do not really support the publication of the paper. Therefore, I encourage the authors to take the comments and suggestions from the reviewers and improve it in order to resubmit to the next conference.


**Award:**

No

---

### Decision · Program_Chairs · 2022-09-14

Reject